# Extravascular optical coherence tomography of cerebral vessel walls in vivo

**Karl Hartmann** [1] *, **Belal Neyazi** [1], **Claudia A. Dumitru** [1], **Aiden Haghikia** [2], **I. Erol Sandalcioglu** [1], **Klaus-Peter Stein** [1]

1 Universitätsklinik für Neurochirurgie, Otto-von-Guericke-Universität Magdeburg, Sachsen-Anhalt, Magdeburg, Deutschland, 2 Universitätsklinik für Neurologie, Otto-von-Guericke-Universität Magdeburg, Sachsen-Anhalt, Magdeburg, Deutschland

* karl.hartmann@med.ovgu.de

## Abstract

### Purpose

Evaluation of extravascular, microscope integrated OCT (iOCT) as an in vivo imaging modality of cerebral blood vessels and as an intraoperative imaging method.

### Methods

Microscope integrated optical coherence tomography of major cerebral arteries (n = 13) and superficial sylvian veins (n = 5) and one incidental cerebral vasospasm (n = 1) in (n = 10) patients. Post procedural analysis of OCT volume scans, microscopic images and videos during the time of scan as well as measurements of the diameter of vessel walls and its layers with an accuracy of 7.5 μm.

### Results

iOCT was feasible during vascular microsurgical procedures. In all scanned arteries a clear delineation of the physiological three layered vessel wall composition could be achieved. Pathological arteriosclerotic alterations of cerebral artery walls could precisely be demonstrated. Major superficial cortical veins conversely presented a mono layered composition. First in vivo measurements of vascular mean diameters were possible. Cerebral artery walls showed a diameter of 296 μm, tunica externa 78 μm, media 134 μm and interna 84 μm.

### Conclusion

For the first time the microstructural composition of cerebral blood vessels could be illustrated in vivo. Due to an outstanding spatial resolution a clear definition of physiological and pathological characteristics was possible. Therefore, microscope integrated optical coherence tomography holds promise for basic research in the field of cerebrovascular arteriosclerotic diseases and for intraoperative guidance during microvascular surgery.

**Data Availability Statement:** All relevant data are within the paper and its Supporting Information files. A data set which demonstrates exemplary raw OCT volume scans of the first in vivo scans of a

human cerebral internal carotid artery and superficial silvian vein vessel wall characteristics as well as an excel sheet with raw results of vessel wall layer diameters were made freely available to improve reproducibility of the results: https://osf.io/a3gx8/?view_only=649952d8965543349ffc3c90cf8d6aea.

**Funding:** The authors received no specific funding for this work.

**Competing interests:** OptoMedical Technologies GmbH, Maria-Goeppert-Straße 9, 23562 Lübeck, Schleswig Holstein, Deutschland supported the study with free equipment for iOCT. This does not alter the adherence to PLOS ONE policies on sharing data and materials. The authors declare no employment, remunerated consultancy, involvements in patents, products in development or marketed products linked to this company. The authors have nothing further to disclose and no further competing interests.

## Introduction

Imaging of cerebral vessel wall characteristics is fundamental to understand pathophysiological processes of cerebrovascular diseases and for intraoperative guidance during micro neurosurgery. Histology—albeit its' superior and unexcelled resolution—excludes an in vivo approach and is further constrained by numerous process-related artifacts, such as specimen shrinkage, tissue damage and tissue separation [1]. Traditional intraoperative imaging modalities such as ultrasound lack the necessary spatial resolution to delineate vessel wall characteristics as well as the possibility of microscope integration [2–4].

OCT imaging depends on near infrared light. In comparison to visible light–it delivers less energy. It has less electronic excitation potential of molecules and no thermal capacities. It is routinely used in ophthalmology since 30 years. No adverse effects or harming potential at the neuronal cell layer of the retina or in general are described so far [5].

In biological tissue penetrating depth is approx. 4000 μm and therefore suitable for cerebral vessel wall imaging [6]. It shows an exceedingly high spatial resolution ranging from 1–15 μm and therefore allows for identification of structural constituents to the extent of histology [7]. In ex vivo experimental set ups the correlation of intravascular OCT and histological findings of vessel walls could already be demonstrated [8–10]. In this context, extravascular OCT even seemed to exceed intravascular OCT in terms of image quality [11–13]. Physically depending on light, microscope integration is fairly simple if optics of microscopes are optimized for light transmission in near infrared spectral range. It then allows for rapid, real-time, three dimensional scanning of tissue in the field of view of the surgeon [14].

Various publications describe endovascular OCT for accessing coronary arteries during cardiac catheterization. Due to size and limited flexibility of fiber optics the intracranial use of intravascular OCT remains highly restricted. In case of the anterior Circle of Willis the intra carotid artery siphon in particular prevents intracranial passage [8]. Consequently, reports of neuro-endovascular OCT remain limited and mostly subject animal or ex vivo models [8,15]. Only a few case reports address in vivo vessel wall imaging of the infra-tentorial cerebral vasculature [16,17].

Our group recently demonstrated that extravascular OCT was feasible during micro neurosurgical procedures [13,18–20].

Due to these appropriate technical properties as well as the feasibility during micro surgical procedures, we here investigate microscope integrated optical coherence tomography as the first suitable in vivo imaging modality of cerebral vessel wall morphology.

## Materials and methods

### Patients

10 patients (6 = female; 4 = male; mean age 54±19; range 17–77) with indication for supra tentorial, micro neurosurgery were included. A cerebral vasospasm (n = 1), superficial sylvian veins (n = 4) and cerebral arteries (n = 13) were selected for OCT scanning. Indication for craniotomy ranged from brain tumors (n = 4) to cerebral aneurysms (n = 6). All patients gave written informed consent. The study was approved by the local ethics committee (No. 3012–2016). Patient characteristics are listed in **Table 1**.

### Optical coherence tomography

If preparation and dissection of cerebral blood vessels was necessary for the surgical procedure OCT-scans were performed. The surgical microscope HS Hi-R1000G (Haag-Streit Surgical GmbH, Wedel, Germany) with integrated OCT-camera (OptoMedical Technologies GmbH,

**Table 1. Patient characteristics.**

| No. | Age | Sex | Scanned Vessel | Scanned Segment | Indication for Surgery |
|---|---|---|---|---|---|
| 1 | 66 | f | ICA<br>SV | C1 | Tumor extra-axial |
| 2 | 77 | m | MCA | M2 | Tumor extra-axial |
| 3 | 71 | m | ICA<br>SV | C1 | Tumor extra-axial |
| 4 | 17 | f | ICA | C1 | Tumor intra-axial |
| 5 | 75 | m | ACA<br>ICA | A1<br>C1 | Cerebral Aneurysm |
| 6 | 46 | f | ACA | A1 | Cerebral Aneurysm |
| 7 | 58 | f | ICA<br>SV | C2 | Cerebral Aneurysm |
| 8 | 44 | f | ICA<br>MCA | C1<br>M2 | Cerebral Aneurysms |
| 9 | 56 | f | ICA<br>SV | C1 | Cerebral Aneurysms |
| 10 | 29 | m | ICA<br>ICA<br>SV | C1<br>C2 | Cerebral Aneurysms |

Displaying patients' characteristics with definition of age, gender, scanned vessels and indications for neurosurgical interventions. Note that definition of scanned segments was limited by the surgical approach but it was aimed for proximal segments of the anterior Circle of Willis. Abbreviations: Internal carotid artery (ICA), middle cerebral artery (MCA), anterior cerebral artery (ACA), anterior communicating artery (AcomA) and sylvian vein (SV).

Lübeck, Germany) was used. A two axis scanner assured three-dimensional real time OCT scanning of tissue in the field of view. Volume scans—consisting of 30 B scans—each with a lateral scan width of 5–37 mm, an optical window depth of 3.1–4.2 mm and an axial spatial resolution 7.5 μm could be obtained.

OCT scanning was screened live on microscope monitors and via a head up display in the field of view of the surgeon. OCT volume scans, corresponding light microscopic pictures and light microscopic videos were further stored for post procedural analysis. The system was CE (conformity with the European Union guidelines) certified for intraoperative documentation of tissue structures. For detailed description on the setup see [19].

### Image acquisition

The OCT scan site was defined by a senior vascular neurosurgeon with experience in OCT scanning. Focusing on the proximal branches of the anterior Circle of Willis with limitation due to the neurosurgical approach. An orthograde scan-angle and highest microscope magnification was intended. A corresponding light microscopic picture, video and three-dimensional OCT volume scan were recorded and stored with a data set defining OCT characteristics at time of scan.

### Anatomical analysis

Data was sorted according image quality and richness of details. The aspect ratio of OCT scans needed to be adjusted according to microscope magnification during time of scan. This process followed a fixed protocol in every subject. These data sets were screened by experienced surgeons for anatomical analysis.

### Distance measurements

A singular B—scan was selected from a set of volume scans. Anatomical structures were measured parallel to the optic path and under maximum augmentation. The center of the probe was selected to reduce scattering effects. Due to the spatial resolution of 7.5 μm final measurements were rounded to tens. Measuring was not adjusted according the index of refraction. For image processing ImageJ2 and Fiji were used [21,22].

## Results

### Adverse effects

In one 17 yo female patient with resection of an extended right temporal mesial Ganglioglioma (WHO Grade I) the below mentioned vasospasm of an M2 segment occurred. The segment was scanned with OCT after opening of the dura mater. After extended microsurgical dissection of the tumor an intraoperative subarachnoid hemorrhage occurred followed by the below described vasospasm. Other modifications of vascular tone in temporal connection with OCT scans did not occur.

None of the participants demonstrated global adverse effects. No prolonged wake up phase, novel neuronal deficits, headaches or epileptic seizures were clustered.

### Cerebral arteries

In all scanned arteries the physiological three-layer composition of the vessel wall could be delineated. Pathological alterations like arteriosclerotic plaques, calcifications and circular inclusions could be further demonstrated. Light microscopy, corresponding OCT scan and illustration of these physiological and pathological characteristics as displayed by OCT are presented in **Fig 1**. The variety of cerebral vessel wall modifications in 10 exemplary OCT B-scans of 10 different cerebral arteries and patients are shown in **Fig 2,** https://osf.io/a3gx8/.

Vessel wall diameter (VWD) largely varied inter- and intra-individually (absolute VWD: mean 296±125 μm, range 60–520 μm; intima: mean 84±38 μm, range 20–160 μm; media: mean 134±73 μm, range 20–290 μm; externa: mean = 78±43 μm, 20–150 μm). The proportion of singular vessel wall layers in relation to the absolute VWD (100%) is further expressed (intima: mean = 30±9%, range = 16–47%; media: mean = 43±10%, range = 28–60%, externa = 27±11%, range = 11–48%). Vessel wall and vessel wall layer diameters are listed in **Table 2,** https://osf.io/a3gx8/.

### Cerebral veins

Scanned sylvian veins (SV) exhibited a thinner vessel wall (mean 53±23 μm, range 30–80 μm) than cerebral arteries. A stratum-like composition was not recognized. Arteriosclerotic modifications were not displayed. Regarding the vessel lumen, veins presented an intraluminal scattering of light. This phenomenon was absent in cerebral arteries without vasospastic conditions. An exemplary light microscopy, corresponding OCT scan and illustration of physiological, anatomical characteristics as displayed by OCT are presented in **Fig 3,** https://osf.io/a3gx8/.

### Cerebral vasospasm

During preparation an incidental cerebral vasospasm could be scanned. Later analysis demonstrated an intra stenotic constriction of the vessel wall diameter, an enlarged diameter of the tunica media–presumably due to constriction of soft muscular tissue—and a mainly pre- but also post stenotic intra luminal scattering of light. Which we interpreted as an increased optical density due to a turbular flow in the periphery in relation to a laminar flow in the center of the lumen. A bird's eye view of a 3-dimensional OCT volume scan and corresponding OCT B-scans are illustrated in **Fig 4.**

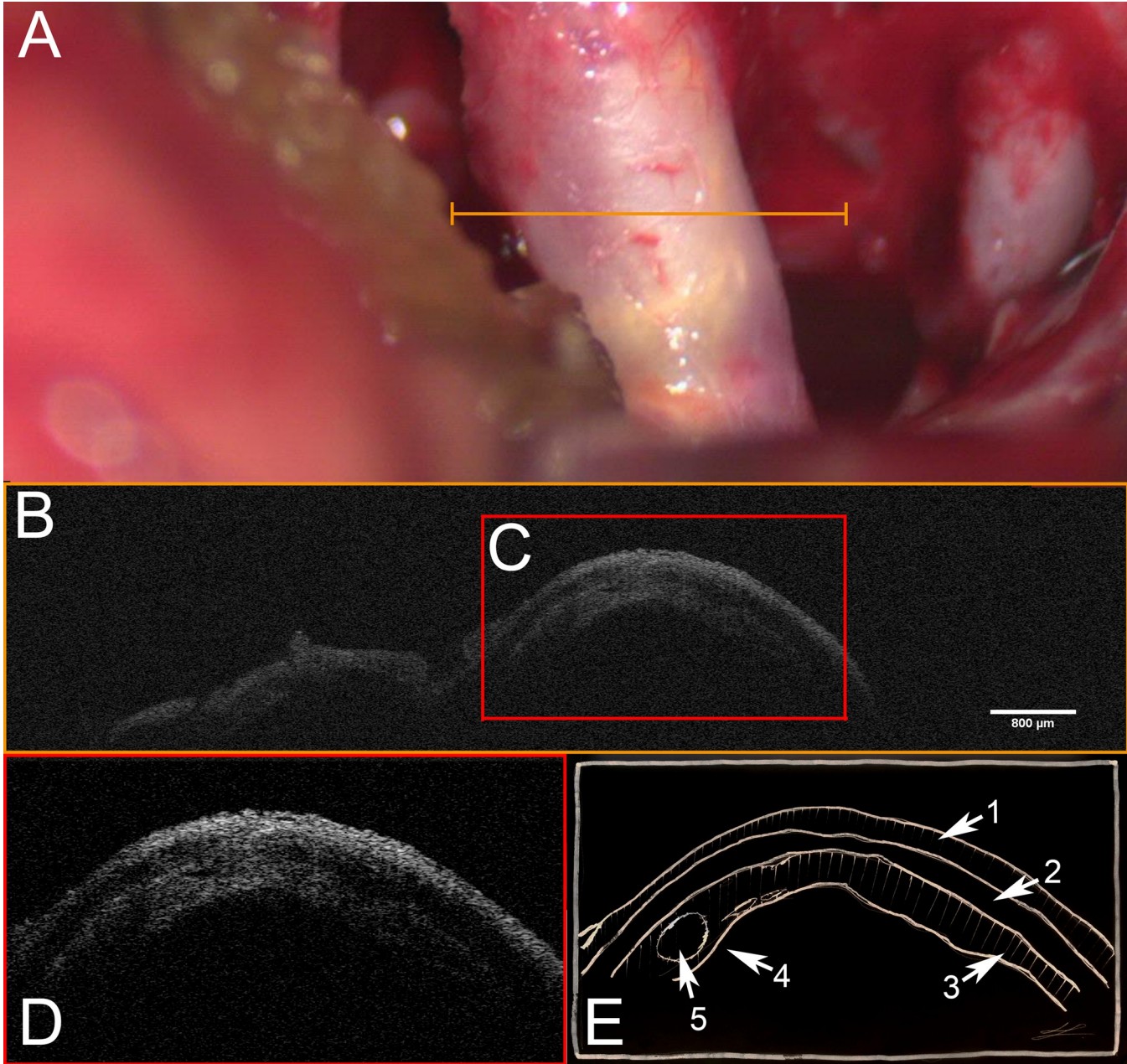

**Fig 1. OCT scan of ICA with arteriosclerotic vessel wall modification. A** Light microscopy of internal carotid artery demonstrating arteriosclerotic modifications. Orange line indicates region of OCT scan. **B** Corresponding OCT-scan. Note the three layered composition of the vessel wall. **C + D** enlarged excerpt of OCT scan. **E** corresponding schematic drawing of microstructures. **1** tunica externa **2** tunica media **3** tunica interna **4** arteriosclerotic segment of vessel wall with intima hyperplasia, increased vessel wall thickness and bulging of vessel wall. **5** circular inclusions could reflect an artheroma. See https://osf.io/a3gx8/ for exemplary raw 3-D OCT scan].

## Discussion

Various attempts using high-field and ultra-high-field MRI addressed non-invasive vessel wall imaging of intracranial cerebral arteries in vivo [23,24]. So far, spatial resolution remains limited with voxel sizes of 500–800 μm inhibiting a detailed depiction of vessel wall composition

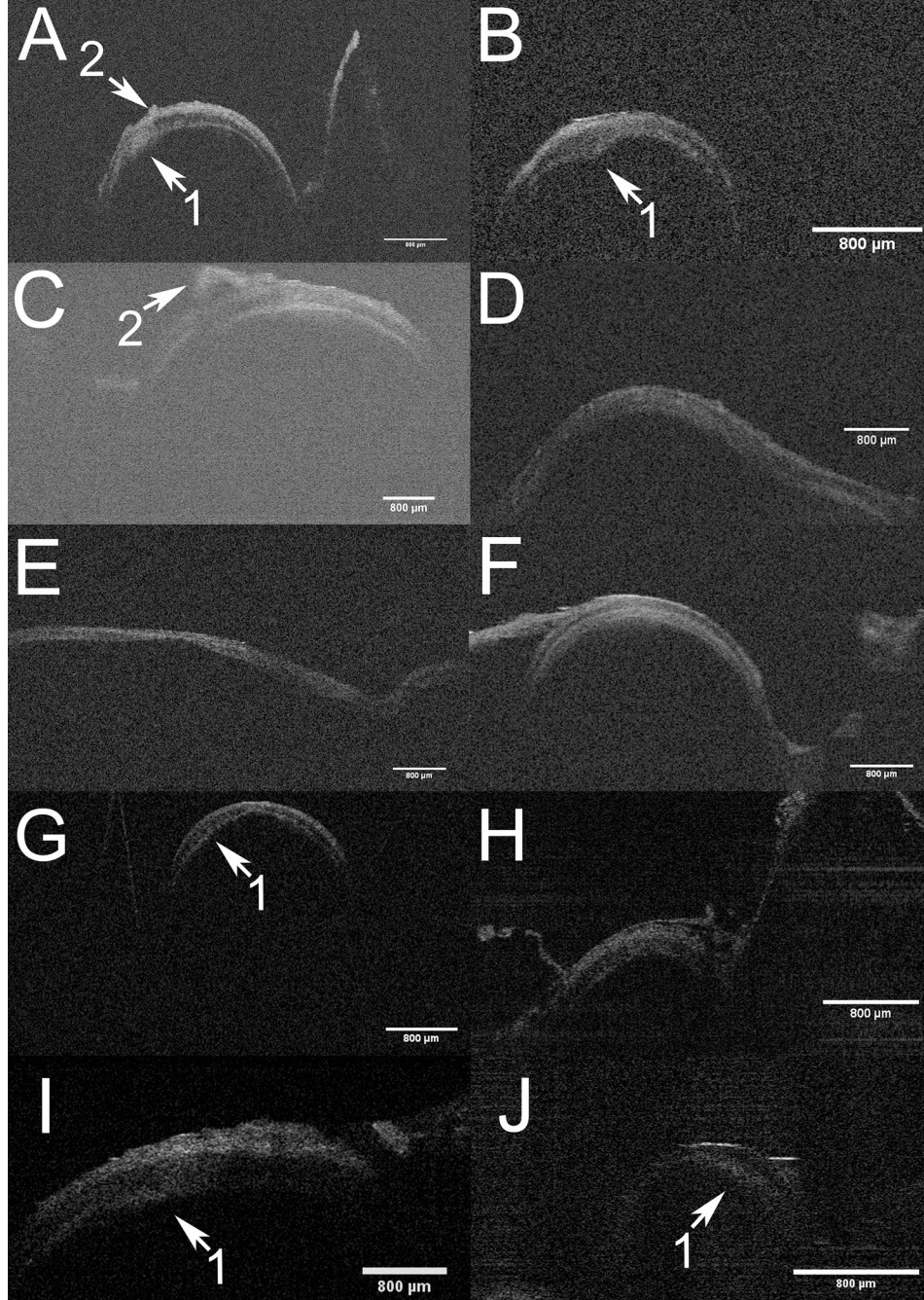

**Fig 2. Variety of cerebral artery vessel wall modifications as displayed by OCT. A-J** Exemplary OCT B-scans of 10 intracranial arteries of the anterior circulation demonstrating the variety of vessel wall modifications as demonstrated by OCT. **E** Shows the appearance of the vessel wall in a longitudinal OCT-scan in contrast to the transverse alignment of other scans. **1** Arteriosclerotic modification of vessel wall layers with increased vessel wall thickness, intima hyperplasia, calcifications and typical "bulging" of vessel wall. **2** Small perforating branches of cerebral arteries as demonstrated by OCT.

[25]. As an intraoperative real-time imaging method ultra sound states the gold standard and is integrated in daily clinical practice. But spatial resolution capabilities also inhibits depiction of the vessel wall composition [2–4].

**Table 2. Vessel wall diameters and proportions of cerebral arteries.**

| No. | Artery | Vessel Wall Diameter absolute in µm | Tunica in µm (% of absolute vessel wall diameter) | | | | | |
|-----|--------|------|-------|-------|-------|-------|-------|-------|
| | | | Intima | | Media | | Externa | |
| 1 | ICA | 60 | 20 | (33) | 20 | (34) | 20 | (33) |
| 2 | ICA | 400 | 120 | (30) | 130 | (33) | 150 | (37) |
| 3 | ICA | 210 | 50 | (24) | 60 | (28) | 100 | (48) |
| 4 | ICA | 300 | 80 | (27) | 180 | (60) | 40 | (13) |
| | ICA | 450 | 130 | (29) | 220 | (49) | 100 | (22) |
| 5 | ICA | 310 | 60 | (20) | 140 | (45) | 110 | (35) |
| 6 | ACA | 370 | 160 | (43) | 140 | (38) | 70 | (19) |
| | ICA | 340 | 90 | (26) | 190 | (56) | 60 | (18) |
| 7 | ACA | 280 | 110 | (39) | 140 | (50) | 30 | (11) |
| 8 | ICA | 120 | 40 | (33) | 50 | (42) | 30 | (25) |
| | ICA | 320 | 50 | (16) | 140 | (44) | 130 | (40) |
| 9 | ICA | 520 | 100 | (19) | 290 | (56) | 130 | (25) |
| 10 | MCA | 170 | 80 | (47) | 50 | (29) | 40 | (24) |
| | **MEAN** | 296±125 | 84±38 | 30±9 | 134±73 | 43±10 | 78±43 | 27±11 |
| | **RANGE** | 60–520 | 20–160 | 16–47 | 20–290 | 28–60 | 20–150 | 11–48 |

Absolute vessel wall diameters of 10 cerebral arteries on the right. Notice the high range of inter-individual variability. Absolute and relative diameters of cerebral vessel wall layers on the right. Mean values and ranges are listed in the bottom rows. Notice the pronounced diameter of tunica media. Notice the wide span in range in all three vessel wall layers reflecting the inter-individual variability. See https://osf.io/a3gx8 for raw data.

In contrast OCT shows an exceptional high spatial resolution—in our set up 7.5 µm—and penetration depth ranges from 1000–3000 µm [15]. In line with this resolution, experimental OCT analysis of vessel walls could priorly demonstrate coherence with histological findings [8–10]. No adverse effects are expected since scanning depends on near infrared light which has less irritation potential then the light spectrum of the operating microscope [26]. Due to its physical properties microscope integration is fairly simple [14]. These properties seem to be suitable to visualize and analyze the microstructural composition of vessel walls and integration of the technique in the microsurgical work flow.

This study presents for the first time detailed human vessel wall visualization of supra-tentorial cerebral vessels in vivo and real time. Due to the size and cross sectional design validity remains limited but no adverse effects were documented.

## Cerebral arteries

OCT scanning revealed a three layered composition in all scanned arteries. Prior studies in coronary OCT verified these optical layers to match histological vessel wall strata [27]. Spatial resolution of OCT scanning met histology at intermediate magnification and imaging was well comprehensible. Pathological changes like arteriosclerotic plaques, calcification or intima hyperplasia could further be visualized and matched to the particular strata. Image quality seemed to outperform those of endovascular approaches [15,16]. In coherence to the literature we postulate that an increased optical density of blood in relation to air lead to increased light absorption during endovascular scanning decreasing image quality [15].

## Vessel wall diameters

To the best of our knowledge, we here present the first in vivo measurements of cerebral vessel wall and single layer diameters. Absolute as well as proportional measurements varied widely.

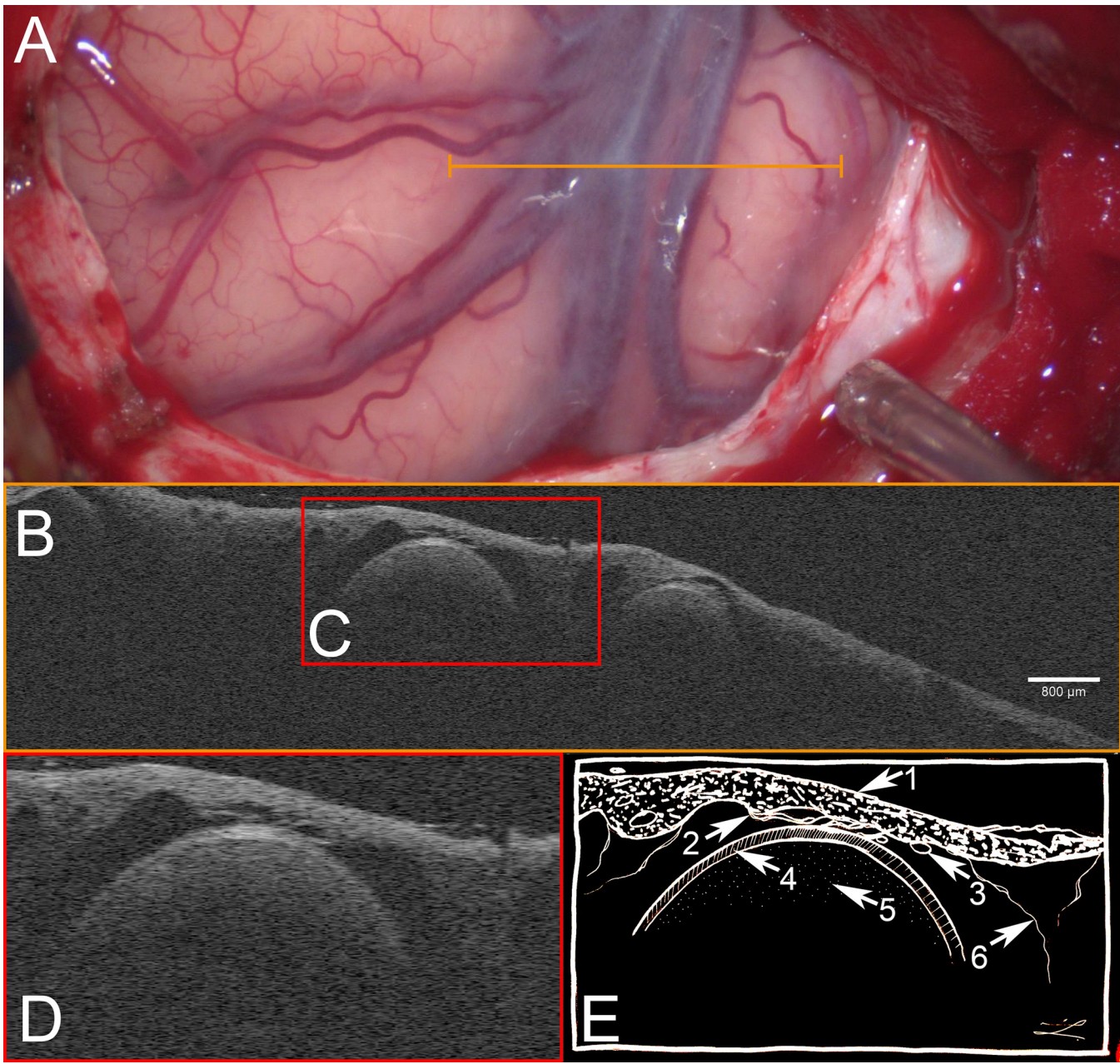

**Fig 3. Trans arachnoid microscope integrated OCT scan of sylvian veins. A** light microscopic image of Sylvian fissure after right fronto lateral craniotomy. Opened segment shows Sylvian fissure with superficial Sylvian veins and temporal as well as frontal brain cortex. Orange line indicates region of scan. **B** OCT-scan of Sylvian veins. **C + D** enlarged excerpt of OCT scan demonstrating characteristics of OCT scans of the Sylvian fissure. **E** schematic drawing of microstructures as depicted by OCT: **1** thick arachnoid barrier cell membrane, **2** fading transition of arachnoid barrier cell membrane to trabecular system, **3** arachnoid blood vessels, **4** vessel wall of Sylvian vein. Note mono layered composition in OCT scan, **5** intraluminal scattering of light, **6** trabecular system. See https://osf.io/a3gx8/ for raw 3-D OCT scan.

Physiological, pathological and methodological factors might influence these measurements. Physiologically we are looking at individual variants of vascular size as well as individual vessel wall layer diameters. Furthermore, we are scanning different vascular tones. Vascular tone not only influences the absolute diameter of vessels but also the diameter of its three tunicae and the proportions among them.

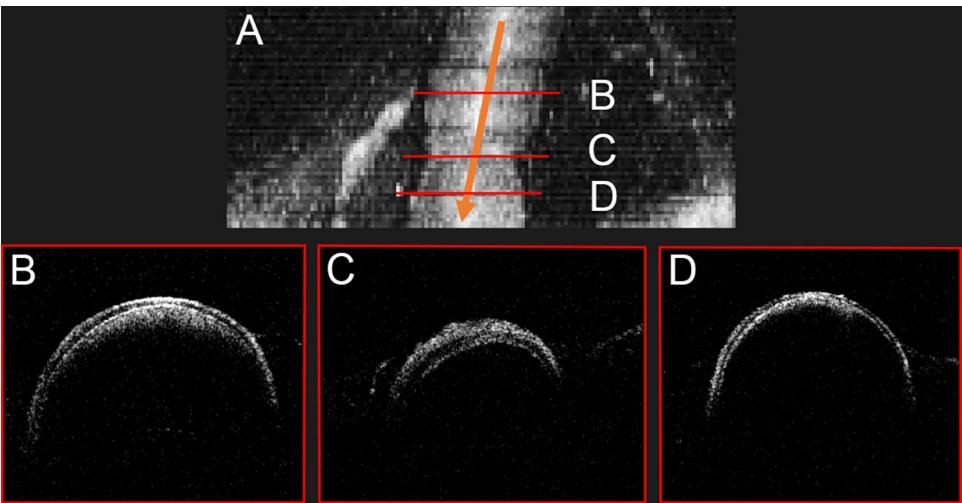

**Fig 4. OCT scan of arterial vasospasm of MCA M2 segment. A-C** Bird's eye view of 3-dimensional OCT volume scan of cerebral artery vasospasm; MCA, M2 segment. Orange arrow demonstrates blood flow direction; red lines represent the position of sectional OCT scans. **D** pre-, **E** intra- and **F** post-stenotic section. The **E** intra-stenotic reduction of vessel diameter is interpreted as local vasoconstriction caused by contracted smooth muscle cells of tunica media. This is illustrated by an increased diameter of tunica media in the **E** intra-stenotic segment in comparison to the **D** pre- and **F** post-stenotic segments. The mainly **D** pre- and also **F** post-stenotic intraluminal scattering of light could be a result of turbulent blood flow in comparison to a laminar flow in the center of the blood vessel.

This physiological mechanism was remarkably illustrated by scanning of an incidental intraoperative vasospasm, see **Fig 4**. Though this adaption of vascular tone occurred after OCT scanning, we do not propagate a causal association. The vasospasm occurred (A) about one hour after the scan, (B) after extended manipulation of the vessel, (C) after iatrogenic sub-arachnoid hemorrhage and (C) we did not see such phenomena in any other scanned vessel though we screened explicitly for vascular layer diameters.

Pathological factors like arteriosclerosis lead to an increased diameter of tunica intima and in late stages to a diminished diameter of tunica media due to muscular atrophy influencing the vessel wall and vessel wall layer diameters [28,29]. This variability of pathological alterations is well demonstrated in **Fig 2**. Methodological factors could be slight deviation from an orthograde scan angel due to the limited surgical approach and—of course—the limited case number.

## Cerebral veins

OCT scans of cerebral veins did not demonstrate a three-layer composition. The minor role of tunica media in veins regarding thickness, structure and function could be accountable. In arteries, this particular layer predominantly allowed to distinguish certain vessel wall layers. Rather thin walls in cerebral veins could further account for these findings. OCT scanning revealed venous wall diameters of approx. 30 μm. Which approaches our methodological limits.

Intraluminal light scattering was only found in veins and not in arteries. The lower blood velocity within veins could promote peripheral turbulent flow, according to Hagen-Poisseuille equation. Turbulent flow could lead to increased light absorption in relation to a central lami-nar flow. This would be in coherence with assumptions based on fluid dynamics and our observations in the vasospastic M2 artery segment [30,31]. Here, scattering of light was most prominent in the pre-stenotic segment, not visible in the intra-stenotic segment and presented again in the post-stenotic segment, see **Fig 4**.

## Future perspectives

These results demonstrate that OCT can be used to study physiological and pathological alterations of vessel walls in vivo. One clinical application would be the field of vasa vasorum. These vessels are crucial in the pathogenesis of arteriosclerosis among multiple other cerebrovascular diseases [32]. Technically, OCT systems with further improved spatial resolution yet exist which could be able to delineate these vessels and adjacent vascular segments from an extravascular approach [33].

Microscope-integrated extravascular OCT should be further valued as an intraoperative imaging tool in the context of plastic, cardiac and neurosurgery. The highly demanding field of vascular neurosurgery with pathologies ranging from cerebral aneurysms, cerebral as well as spinal arterio-venous-malformations and Moyamoya disease could be targeted among others. In the latter, OCT could explicitly delineate the optimal site for vascular harvest and extra-intra cranial bypass [34].

## Conclusion

We here present the first in vivo imaging and measurements of cerebral vessel walls with extravascular microscope integrated OCT. Clinical adverse effects were not noted. Other optical based instruments of the neurosurgical operation room like the navigation system, infrared angiography or electro physiological measurements seemed not to be influenced.

It allowed detailed depiction of single arterial vessel wall layers, approaching spatial resolution of histopathology. Physiological changes of vascular tone with consecutive modification of the vascular lumen as well as diameters of singular arterial layers could be demonstrated clearly and in real time. Pathological conditions like arteriosclerotic modifications with intima hyperplasia, calcifications and local changes of vessel wall diameters could be depicted. Future studies should value microscope integrated OCT for basic research and as an intraoperative guidance tool in vascular surgery.

## Author Contributions

**Conceptualization:** Karl Hartmann, Belal Neyazi, I. Erol Sandalcioglu, Klaus-Peter Stein.

**Data curation:** Karl Hartmann, Klaus-Peter Stein.

**Formal analysis:** Karl Hartmann, Klaus-Peter Stein.

**Investigation:** Belal Neyazi.

**Methodology:** Karl Hartmann.

**Supervision:** I. Erol Sandalcioglu.

**Validation:** Karl Hartmann, I. Erol Sandalcioglu, Klaus-Peter Stein.

**Visualization:** Klaus-Peter Stein.

**Writing – original draft:** Karl Hartmann, Klaus-Peter Stein.

**Writing – review & editing:** Karl Hartmann, Claudia A. Dumitru, Aiden Haghikia, Klaus-Peter Stein.

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
