## [Decision Letter · Decision Letter 0]

6 Jun 2022

PONE-D-21-23756Optical Coherence Tomography of Cerebral Vessel Walls in VivoPLOS ONE

Dear Dr. Hartmann,

Thank you for submitting your manuscript to PLOS ONE. After careful consideration, we feel that it has merit but does not fully meet PLOS ONE’s publication criteria as it currently stands. Therefore, we invite you to submit a revised version of the manuscript that addresses the points raised during the review process.

We look forward to receiving your revised manuscript.

Kind regards,

Alfred Pokmeng See, M.D.

Academic Editor

PLOS ONE

Journal Requirements:

2. Thank you for stating the following financial disclosure: "No funding was received."

3. Thank you for stating the following in the Competing Interests section: "The authors have nothing to disclose and no competing interests. OptoMedical Technologies GmbH supported the study with free equipment for iOCT." 

We note that you received funding from a commercial source: [Name of Company]

5. Please amend the manuscript submission data (via Edit Submission) to include author I. Erol Sandalcioglu.

Additional Editor Comments (if provided):

Dr. Hartmann, your group is performing important work in advancing the ability of neurosurgeons to evaluate vessel structure in the operating room, which could improve diagnosis and management of intrinsic cerebrovascular disease and iatrogenic cerebrovascular pathology. The continued publication of baseline/normative data and pathologic variation is important.

The reviewers comment on a number of critical ways in which this manuscript could improve the communication and documentation of this data to advance the field. I recommend further analysis of data and corrections as they suggest.

Reviewers' comments:

Reviewer's Responses to Questions

**Comments to the Author**

1. Is the manuscript technically sound, and do the data support the conclusions?

Reviewer #1: Yes

Reviewer #2: No

2. Has the statistical analysis been performed appropriately and rigorously? 

Reviewer #1: N/A

Reviewer #2: No

3. Have the authors made all data underlying the findings in their manuscript fully available?

Reviewer #1: Yes

Reviewer #2: Yes

4. Is the manuscript presented in an intelligible fashion and written in standard English?

Reviewer #1: Yes

Reviewer #2: Yes

5. Review Comments to the Author

Reviewer #1: In the present manuscript, Hartman et al apply in vivo extravascular microscope integrated OCT (iOCT) of the cerebral vessel wall during neurosurgical procedures to analyze cerebral vessel wall layers of both arteries and veins. Supported by illustrative figures, they show that iOCT is a feasible technique. They furthermore provide measures of different cerebral vessel wall layers in vivo for the first time. The topic is of relevance for the field and is both interesting and relevant for neurologists and neurosurgeons since the technique of iOCT might allow to address open pathophysiological questions of cerebral vessel diseases. The methods are described in detail and are sound. The results are presented adequately and support the derived conclusions.

I have one major and some minor comments which I hope the authors can address in a revise version of the manuscript.

MAJOR:

Results: Please describe whether any complications or safety issues occurred during iOCT analysis. Since iOCT is a medical procedure involving humans, addressing its safety and risk profile is crucial.

MINOR:

Methods: To allow reproducibility of the findings, the authors should describe in detail where they exactly analyzed the respective vessels. For example, regarding MCA/ACA/ICA, the authors may provide which segments have been measured by iOCT. This information might be included into table 1.

Results: In Table 1, the authors used the abbreviations “ICA” and “ACI”, whereas “ACI” is not defined within the table legend. I suggest that ACI might also reflect ICA, please clarify and revise accordingly. Please use the same abbreviations for the same vessel structures in all provided tables and figures (table 1: ICA, table 2: ACI).

Results: I have doubts whether it is useful to provide a mean vessel diameter in table 2 which is derived from three different anatomical vessel structures (ACI, ACA, MCA). I suggest providing mean vessel diameters of specific cerebral vessels that were analyzed in comparable segments and regions (see also above).

Results: Did the authors recognize any associations of their iOCT findings with age and sex of the respective patients?

Discussion: The authors claim to provide the first in vivo measures of cerebral vessel layers derived from OCT-A. Please discuss whether the findings from iOCT correspond to histological studies of comparable vessels and comparable regions.

Discussion: Future perspectives: From my point of view, there are several more fields that could be addressed by iOCT. For example, pathophysiological features and findings in cerebral aneurysms (especially after successful clipping) would be very favorable.

Reviewer #2: The authors use of microscope integrated optical coherence tomography (OCT) is similar to the groups prior published papers. Their prior publications show that OCT can be successfully used in-vivo to demonstrate baseline characteristics of a particular pathology such as an aneurysm or an arachnoid cyst. Outside of the novel use of this application, which they have already demonstrated, it is difficult to ascertain what they are trying to show with their data. There is already limited information in the presented patient characteristics, and no attempt to demonstrate consistency within the presented factors. Even the comparison of vessel wall and vessel wall layers do not all compare the same type of arteries, and their only reasonable conclusion is that artery sizes vary. Overall, the authors show this particular use of OCT is possible, which has already been demonstrated, but fail to bring any new information regarding this technique.

6. PLOS authors have the option to publish the peer review history of their article (what does this mean?). If published, this will include your full peer review and any attached files.

Reviewer #1: **Yes: **PD Dr. Benjamin Knier

Reviewer #2: No

---

## [Author Response · Author response to Decision Letter 0]

20 Jul 2022

Comments to the 1st Reviewer

Reviewer #1: In the present manuscript, Hartman et al apply in vivo extravascular microscope integrated OCT (iOCT) of the cerebral vessel wall during neurosurgical procedures to analyze cerebral vessel wall layers of both arteries and veins. Supported by illustrative figures, they show that iOCT is a feasible technique. They furthermore provide measures of different cerebral vessel wall layers in vivo for the first time. The topic is of relevance for the field and is both interesting and relevant for neurologists and neurosurgeons since the technique of iOCT might allow to address open pathophysiological questions of cerebral vessel diseases. The methods are described in detail and are sound. The results are presented adequately and support the derived conclusions.

Response: We appreciate the positive evaluation of the manuscript.

I have one major and some minor comments which I hope the authors can address in a revise version of the manuscript.

MAJOR:

Results: Please describe whether any complications or safety issues occurred during iOCT analysis. Since iOCT is a medical procedure involving humans, addressing its safety and risk profile is crucial.

Response: OCT imaging depends on backscattered infrared light. Its electromagnetic energy is lower than the one of visual light. Therefore, its capability for electronic excitation of molecules also remains lower. Leading to lesser tissue irritation then visual light. We now clarified this in the method section:

OCT imaging depends on near infrared light resulting in less electronic excitation then the visible light spectrum and is therefore harmless to biological tissue5. In biological tissue penetrating depth is approx. 4000 μm and therefore suitable for cerebral vessel wall imaging7.

MINOR:

Methods: To allow reproducibility of the findings, the authors should describe in detail where they exactly analyzed the respective vessels. For example, regarding MCA/ACA/ICA, the authors may provide which segments have been measured by iOCT. This information might be included into table 1.

Response: We appreciate this suggestion which can improve the accuracy of the manuscript. Table 1 and the Method section has been amended accordingly, see "Revised Manuscript with Track Changes".

Response: Both tables were revised as suggested by the reviewer. See "Revised Manuscript with Track Changes":

Results: I have doubts whether it is useful to provide a mean vessel diameter in table 2 which is derived from three different anatomical vessel structures (ACI, ACA, MCA). I suggest providing mean vessel diameters of specific cerebral vessels that were analyzed in comparable segments and regions (see also above).

Response: We see the point of the reviewer and appreciate the suggestion. But detailed analysis of our primary data showed no consistency or trends of vessel wall diameters in relation to anatomical parts of the cerebral vasculature e. g. C1 segment of ICA (see Supp. Inf. Excel sheet). Therefore - and because of the limited number of measured vessel walls in general - we decided to display the absolute results in Table 2.

Results: Did the authors recognize any associations of their iOCT findings with age and sex of the respective patients?

Response: The limited number of participants inhibits a statistical significant analysis of such correlations. Therefore, we would like to exclude such statements from the manuscript. But also descriptively during data analysis such tendencies were not observed.

Discussion: The authors claim to provide the first in vivo measures of cerebral vessel layers derived from OCT-A. Please discuss whether the findings from iOCT correspond to histological studies of comparable vessels and comparable regions.

Response: We thank the reviewer for this suggestion to improve our manuscript. We now provide more details on the correlation between histological studies and OCT in the “Introduction” and “Discussion” sections, respectively:

Imaging of cerebral vessel wall characteristics is fundamental to understand pathophysiological processes of cerebrovascular diseases and for intraoperative guidance during micro neurosurgery. Histology - albeit its’ superior and unexcelled resolution - excludes an in vivo approach and is further constrained by numerous process-related artifacts, such as specimen shrinkage, tissue damage and tissue separation1

It shows an exceedingly high spatial resolution ranging from 1 - 15 µm and therefore allows for identification of structural constituents to the extent of histology7. In ex vivo experimental set ups the correlation of intravascular OCT and histological findings of vessel walls could already be demonstrated 8 9 10. In this context, extravascular OCT even seemed to exceed intravascular OCT in terms of image quality 11 12 13. Physically depending on light, microscope integration is fairly simple…

exceptional high spatial resolution - in our set up 7.5 µm - and penetration depth ranges from 1000 – 3000 µm16. In line with this resolution experimental OCT analysis of vessel walls could priorly demonstrate coherence with histological findings8 9 10. ... 

Discussion: Future perspectives: From my point of view, there are several more fields that could be addressed by iOCT. For example, pathophysiological features and findings in cerebral aneurysms (especially after successful clipping) would be very favorable.

Response: With this manuscript we would like to focus primarily on the possibilities of the technique according fundamental research on dynamic microanatomy in vivo and pathological alterations of such like arteriosclerosis.

But we also see the value of the technique in several domains of microvascular surgery like aneurysm, bypass, arteriovenous malformation as well as cranial and spinal arteriovenous fistula surgeries. We therefore expanded the “Future Perspectives” section of the manuscript:

Microscope-integrated extravascular OCT should be further valued as an intraoperative imaging tool in the context of cardiac and neurosurgery. The highly demanding field of vascular neurosurgery with pathologies ranging from cerebral aneurysms, cerebral as well as spinal arterio-venous-malformations and Moyamoya disease could be targeted among others. In the latter, OCT could explicitly delineate the optimal site for vascular harvest and extra-intra cranial bypass 34. 

\fComments to the 2nd Reviewer

Reviewer #2: The authors use of microscope integrated optical coherence tomography (OCT) is similar to the groups prior published papers. Their prior publications show that OCT can be successfully used in-vivo to demonstrate baseline characteristics of a particular pathology such as an aneurysm or an arachnoid cyst. Outside of the novel use of this application, which they have already demonstrated, it is difficult to ascertain what they are trying to show with their data. 

Response: The results presented in this study are the first extravascular OCT scans and measurements of cerebral vessel wall layers in vivo – and for this reason alone they need to be reported. The application of OCT in this domain – in comparison to other neurosurgical domains – is unique since it demonstrates the (a) value of OCT as a fundamental research tool for microanatomy and clinical highly relevant pathologies of vessel walls e. g. arteriosclerosis, (b) the variability of vessel wall layer diameters among individuals, (c) the variability of arteriosclerotic alterations of cerebral vessel walls, (d) the variation of endo-luminal diameters and diameters of single cerebral vessel wall layers in cerebral vasospasm and (e) the possibility of visualizing turbulent blood flow. These five points are demonstrated in the manuscript.

There is already limited information in the presented patient characteristics, and no attempt to demonstrate consistency within the presented factors. Even the comparison of vessel wall and vessel wall layers do not all compare the same type of arteries, 

Response: A detailed range of patient characteristics were analyzed. But relevant tendencies were not observed and therefore not reported. Furthermore, the number of participants itself inhibits relevant statistical analysis.

Naturally, only vessels which were exposed to the individual surgical approach could be scanned, leading to a certain degree of variability. But the surgeons focused on proximal branches of the anterior Circle of Willis to ensure comparability. This is now explained in more detail in the manuscript, see below.

We also appreciate the remark about vessel segment specification. Therefore, Table 1 and its header were edited to demonstrate the scanned segments. This highlights that mainly the C1 Segment of ICA and near proximal branches were scanned outlining the comparability of vessel wall diameter measurements:

... The OCT scan site was defined by a senior vascular neurosurgeon with experience in OCT scanning. Focusing on the proximal branches of the anterior Circle of Willis with limitation due to the neurosurgical approach. An orthograde scan-angle and highest microscope magnification was intended... 

and their only reasonable conclusion is that artery sizes vary.

Response: In the discussion we pointed out physiological, pathophysiological as well as methodological factors which might influence the variability of vessel wall and vessel wall layer diameters. We now edited this paragraph for more clearness:

To the best of our knowledge, we here present the first in vivo measurements of cerebral vessel wall and single layer diameters. Absolute as well as proportional measurements varied widely. Physiological, pathological and methodological factors might influence these measurements. Physiologically we are looking at individual variants of vascular size as well as individual vessel wall layer diameters. Furthermore, we are scanning different vascular tones. Vascular tone not only influences the absolute diameter of vessels but also the diameter of its three tunicae and the proportions among them. This physiological mechanism was remarkably illustrated by scanning of an incidental intraoperative vasospasm, see Fig 4. Pathological factors like arteriosclerosis lead to an increased diameter of tunica intima and in late stages to a diminished diameter of tunica media due to muscular atrophy influencing the vessel wall and vessel wall layer diameters 29 30. This variability of pathological alterations is well demonstrated in Fig 2. Methodological factors could be slight deviation from an orthograde scan angel due to the limited surgical approach and - of course - the limited case number. 

Overall, the authors show this particular use of OCT is possible, which has already been demonstrated, but fail to bring any new information regarding this technique.

Response: The applicability of microscope integrated OCT in neurosurgical and specific vascular neurosurgical settings was demonstrated elsewhere in detail (Böhringer 2009, Lankenau 2013, Hartmann 2019). This manuscript focuses instead on the validation of this imaging technique for cerebral vessel walls and on the discussion of these results.

---

## [Decision Letter · Decision Letter 1]

25 Aug 2022

PONE-D-21-23756R1Extravascular Optical Coherence Tomography of Cerebral Vessel Walls in VivoPLOS ONE

Dear Dr. Hartmann,

Thank you for submitting your manuscript to PLOS ONE. After careful consideration, we feel that it has merit but does not fully meet PLOS ONE’s publication criteria as it currently stands. Therefore, we invite you to submit a revised version of the manuscript that addresses the points raised during the review process.

We look forward to receiving your revised manuscript.

Kind regards,

Alfred Pokmeng See, M.D.

Academic Editor

PLOS ONE

Journal Requirements:

Additional Editor Comments:

Reviewer 1 is concerned about the lack of reporting of procedural complications associated with OCT via microscopy. Although the authors describe one technical aspect of OCT imaging, regarding light excitation of tissue, there may be both expected and unexpected adverse events with any medical modality. Thermal tissue injury represents one type of risk. Perhaps the authors may explicitly state parameters of excitation associated with OCT optics, in comparison with those of xenon/other visual light microscopy, NIRS probes, and versus ablation lasers used in the operating room.

It may also be appropriate to comment on any observed vasospasm during the cases.

Can the authors comment on interference with other medical devices? For example, optical-based navigation systems can interfere with pulse oximetry and induce artifacts. Both of these other medical technologies could be impacted by OCT light scatter.

Reviewers' comments:

Reviewer's Responses to Questions

**Comments to the Author**

1. If the authors have adequately addressed your comments raised in a previous round of review and you feel that this manuscript is now acceptable for publication, you may indicate that here to bypass the “Comments to the Author” section, enter your conflict of interest statement in the “Confidential to Editor” section, and submit your "Accept" recommendation.

Reviewer #1: (No Response)

2. Is the manuscript technically sound, and do the data support the conclusions?

Reviewer #1: Yes

3. Has the statistical analysis been performed appropriately and rigorously? 

Reviewer #1: Yes

4. Have the authors made all data underlying the findings in their manuscript fully available?

Reviewer #1: Yes

5. Is the manuscript presented in an intelligible fashion and written in standard English?

Reviewer #1: Yes

6. Review Comments to the Author

Reviewer #1: The authors have sufficiently addressed all of my minor concerns. The major concern, however, has not been adequately addressed. Please clearly and explicitly state within the results section, that no or which complications or safety issues occurred during iOCT analysis. iOCT is an invasive procedure and thus addressing its safety and risk profile is crucial. Besides this, I have no further comments.

7. PLOS authors have the option to publish the peer review history of their article (what does this mean?). If published, this will include your full peer review and any attached files.

Reviewer #1: **Yes: **PD Dr. Benjamin Knier

---

## [Author Response · Author response to Decision Letter 1]

8 Sep 2022

Dear PD Dr med Benjamin Knier,

enclosed is the 2nd revision of our manuscript, entitled

"Extravascular Optical Coherence Tomography of Cerebral Vessel Walls in Vivo"

 (PONE-D-21-23756)

Your major concern was indeed not addressed sufficiently. In this regard we further adopted the introduction, results and discussion section of the manuscript. The comments helped us to further reflect the scope of our study and the value of our results in the context of adverse effects of OCT as an intraoperative neuroimaging modality as well as possible interferences with other optical based operating room equipment.

We further edited the reference list.

Edits to the main manuscript have been highlighted in yellow in the document “Revised Manuscript with Track Changes”. None of the figures were changed. 

We are looking forward to your positive consideration.

Yours sincerely, on behalf of the authors,

Karl Hartmann

---

## [Decision Letter · Decision Letter 2]

5 Oct 2022

Extravascular Optical Coherence Tomography of Cerebral Vessel Walls in Vivo

PONE-D-21-23756R2

Dear Dr. Hartmann,

We’re pleased to inform you that your manuscript has been judged scientifically suitable for publication and will be formally accepted for publication once it meets all outstanding technical requirements.

Kind regards,

Alfred Pokmeng See, M.D.

Academic Editor

PLOS ONE

Reviewers' comments:

Reviewer's Responses to Questions

**Comments to the Author**

1. If the authors have adequately addressed your comments raised in a previous round of review and you feel that this manuscript is now acceptable for publication, you may indicate that here to bypass the “Comments to the Author” section, enter your conflict of interest statement in the “Confidential to Editor” section, and submit your "Accept" recommendation.

Reviewer #1: All comments have been addressed

2. Is the manuscript technically sound, and do the data support the conclusions?

Reviewer #1: Yes

3. Has the statistical analysis been performed appropriately and rigorously? 

Reviewer #1: Yes

4. Have the authors made all data underlying the findings in their manuscript fully available?

Reviewer #1: Yes

5. Is the manuscript presented in an intelligible fashion and written in standard English?

Reviewer #1: Yes

6. Review Comments to the Author

Reviewer #1: (No Response)

7. PLOS authors have the option to publish the peer review history of their article (what does this mean?). If published, this will include your full peer review and any attached files.

Reviewer #1: **Yes: **PD Dr. Benjamin Knier

---

## [Editor Report · Acceptance letter]

1 Dec 2022

PONE-D-21-23756R2 

Extravascular Optical Coherence Tomography of Cerebral Vessel Walls in Vivo 

Dear Dr. Hartmann:

I'm pleased to inform you that your manuscript has been deemed suitable for publication in PLOS ONE. Congratulations! Your manuscript is now with our production department. 

Kind regards, 

on behalf of

Dr. Alfred Pokmeng See 

Academic Editor

PLOS ONE